# Limited value of neutrophil-to-lymphocyte ratio and serum creatinine as point-of-care biomarkers of disease severity and infection mortality in patients hospitalized with COVID-19

**Abdisa Tufa**[1]*, **Tewodros Haile Gebremariam**[2], **Tsegahun Manyazewal**[3], **Yidnekachew Asrat**[2], **Tewodros Getinet**[4], **Tsegaye Gebreyes Hundie**[5], **Dominic-Luc Webb**[6], **Per M. Hellström**[6], **Solomon Genet**[1]

1 Department of Medical Biochemistry, School of Medicine, College of Health Sciences, Addis Ababa University, Addis Ababa, Ethiopia, 2 Department of Internal Medicine, School of Medicine, College of Health Sciences, Addis Ababa University, Addis Ababa, Ethiopia, 3 Centre for Innovative Drug Development and Therapeutic Trials for Africa (CDT-Africa), College of Health Sciences, Addis Ababa University, Addis Ababa, Ethiopia, 4 School of Public Health, Saint Paul's Hospital Millennium Medical College, Addis Ababa, Ethiopia, 5 Eka Kotebe General Hospital, Addis Ababa, Ethiopia, 6 Gastroenterology and Hepatology Unit, Department of Medical Sciences, Uppsala University, Uppsala, Sweden

* abdisa.tufa@aau.edu.et

## Abstract

### Introduction

In hospitalized COVID-19, neutrophil-to-lymphocyte ratio (NLR) and serum creatinine is sometimes measured under assumption they predict disease severity and mortality. We determined the potential value of NLR and serum creatinine as predictors of disease severity and mortality in COVID-19.

### Methods

Prospective cohort study of COVID-19 patients admitted to premier COVID-19 treatment hospitals in Ethiopia. Predictive capability of biomarkers in progression and prognosis of COVID-19 was analyzed using receiver operating characteristics. Survival of COVID-19 patients with different biomarker levels was computed. Logistic regression assessed associations between disease severity and mortality on NLR and serum creatinine adjusted for odds ratio (AOR).

### Results

The study enrolled 126 adults with severe (n = 68) or mild/moderate (n = 58) COVID-19, with median age 50 [interquartile range (IQR 20–86)]; 57.1% males. The NLR value was significantly higher in severe cases [6.68 (IQR 3.03–12.21)] compared to the mild/moderate [3.23 (IQR 2.09–5.39)], with the NLR value markedly associated with disease severity (*p<0.001*). Mortality was higher in severe cases [13 (19.1%)] compared to mild/moderate

**Data Availability Statement:** All relevant data are within the paper and its Supporting Information files.

**Funding:** No authors received funding for this particular study. Centre for Innovative Drug Development and Therapeutic Trials for Africa (CDT-Africa), College of Health Sciences, Addis Ababa University, Addis Ababa supported expenses associated with data and sample collection. Professor Per M. Hellström and Associate professor Dominic-Luc Webb, Gastroenterology and Hepatology Unit, Department of Medical Sciences, Uppsala University, Sweden support all expenses related to laboratory analysis. The funders have no role in the study design, data and decision to publish including preparation of the manuscript.

**Competing interests:** The authors have declared that no competing interests exist.

cases [2 (3.4%)] ($p = 0.007$). The NLR value was significantly higher in non-survivors [15.17 (IQR 5.13–22.5)] compared to survivors [4.26 (IQR 2.40–7.90)] ($p = 0.002$). Serum creatinine was significantly elevated in severe cases [34 (50%)] compared with mild/moderate [11 (19%)] ($p<0.001$). Disease severity [AOR 6.58, 95%CI (1.29–33.56), $p = 0.023$] and NLR [AOR 1.07, 95%CI (1.02–1.12), $p = 0.004$)] might be associated with death. NLR had a sensitivity and specificity of 69.1% and 60.3% as predictor of disease severity (cut-off >4.08), and 86.7% and 55.9% as prognostic marker of mortality (cut-off >4.63).

## Conclusion

In COVID-19, NLR is a biomarker with only modest accuracy for predicting disease severity and mortality. Still, patients with NLR >4.63 are more likely to die. Monitoring of this biomarker at the earliest stage of the disease may predict outcome. Additionally, high creatinine seems related to disease severity and mortality.

## 1. Introduction

Coronavirus disease 2019 (COVID-19) is a highly contagious disease caused by the severe acute respiratory syndrome coronavirus 2 (SARS-CoV-2) and poses a serious public health threat around the world [1]. The most common mode of transmission for SARS-CoV-2 is human-to-human contact, such as inhalation or contact with infected droplets. A fecal-oral route is also supported by evidence [2]. COVID-19 affects people differently, with the majority of cases presenting as a mild disease affecting only the upper respiratory tract. In a few cases, however, it can spread to the lower respiratory tract, leading to acute respiratory distress syndrome (ARDS), respiratory failure, multiple organ dysfunction syndrome and death. In COVID-19, a number of risk variables have been linked to disease severity and mortality [3].

According to studies, COVID-19 non-survivors were more likely to be older males with hyperlipidemia, cardiovascular disease (CVD), diabetes mellitus (DM), hypertension, a history of smoking, and chronic obstructive pulmonary disease (COPD) [4, 5].

A cytokine storm is released during the rapid progression of COVID-19 which impairs the immune system by reducing the lymphocyte numbers, particularly T lymphocytes while the number of neutrophils is elevated and increases the neutrophil-to-lymphocyte ratio (NLR) as an independent biomarker of poor clinical outcome [6, 7]. The SARS-CoV-2 infection causes an immune dysregulation with hyper-inflammation that is highly associated with ARDS involving release of pro-inflammatory cytokines and chemokines [8, 9] influencing the severity of disease and is the main cause of death in COVID-19 patients.

NLR has shown predictive values for disease progression and clinical outcomes in illnesses such as COPD, CVD and pancreatitis [10–12]. Many studies have recently reported on the role of NLR in distinguishing mild/moderate cases of COVID-19 from severe ones. Several studies suggest that NLR can be a valid predictor of COVID-19 progression, with high NLR linked to increased COVID-19 mortality [13–15]. Research has also shown NLR to be a cost-effective biomarker for predictions in COVID-19 [16].

Although much information has been gathered on the clinical characteristics of COVID-19, there is a knowledge gap regarding biomarkers of organ function abnormalities, disease severity and outcome from an Ethiopian context. Our primary aim was to determine the relationship between NLR and severity of illness in COVID-19 patients. As secondary aim we

made an effort to identify aberrant organ function tests in hospitalized patients with SARS--CoV-2 infection.

## 2. Materials and methods

### 2.1. Study design and setting

This prospective cohort study was conducted at Tikur Anbessa Specialized Hospital (TASH) and Eka Kotebe General Hospital (EKGH), both located in Addis Ababa, Ethiopia. TASH is the main hospital of Addis Ababa University (AAU), College of Health Sciences (CHS), School of Medicine (SoM), as one of the COVID-19 diagnosis and treatment centers in Addis Ababa. EKGH is a premier COVID-19 treatment facility with an intensive care unit.

### 2.2. Study subjects

The study participants consisted of 126 cases who were 20 years and older with a positive SARS-CoV-2 test result. Diagnosis of COVID-19 was made by positive reverse transcriptase-polymerase chain reaction (RT-PCR) test for SARS-CoV-2 from nasopharyngeal or oropharyngeal swabs.

The recruited COVID-19 patients were divided into two groups: mild or moderate (n = 58) cases and severe (n = 68) cases.

Participants who were anemic, pregnant, unconsciousness, or had cardiac, renal or hepatic failure were excluded from the study.

### 2.3. Sample collection and clinical chemistry analyses

Study participants were evaluated for eligibility and enrolled in the study following their written consent. Relevant sociodemographic information, details of their current illness, past illness including comorbidities and any substance use patterns were collected using a structured questionnaire.

Five mL of whole blood were collected from an antecubital vein by experienced medical laboratory technologists/scientists. One mL of blood was transferred to EDTA-coated tubes and thoroughly mixed for complete blood count, including white blood cell count (WBC). NLR was calculated by dividing the number of neutrophils by the number of lymphocytes per microliter of whole blood. The remaining 4 mL was transferred into serum separator tubes (SST), allowed to form a clot for 30 minutes and subjected to centrifugation at 2200 rpm for 10 minutes. Then, serum was immediately separated, transferred into cryo tubes and stored in aliquots at -80˚C. The serum was used for clinical chemistry test parameters.

The COBAS 6000 automated clinical chemistry analyzer was used to assess the liver enzymes and renal function assays. Liver affection was analyzed by alanine transaminase (ALT; reference 0–33 U/L), aspartate transaminase (AST, reference 0–35 U/L), and alkaline phosphatase (ALP, reference 45–87 U/L), while creatinine (reference 0.5–0.9 mg/dL) and urea (reference 10–45 mg/dL) were used to monitor renal function. All clinical laboratory tests and interpretations were carried out in accordance with the manufacturer's instructions and standard operating procedures.

### 2.4. Operational definitions

Mild/moderate cases of COVID-19: mild were defined as not being hypoxic and without evidence of pneumonia. Moderate cases had clinical signs of pneumonia (fever, cough) but no signs of severe pneumonia (oxygen saturation ≥90%). Severe cases were admitted to the intensive care unit due to severe hypoxemia (oxygen saturation <90%) [17]. Severity was defined by

the greatest illness level reached throughout hospitalization, from admission to occurrence of outcomes (survivor/non-survivor).

## 2.5. Ethics approval

The institutional review board of the College of Health Sciences, AAU (Meeting 01/2021, Protocol 004/21/Biochem) and the research ethics and review committee of the Department of Biochemistry, CHS, AAU (Ref.No. SoM/BCHM/068/2013) approved the study. The study was also approved by Ministry of Science of Ethiopia and Higher Education's national research ethics review committee (Ref.No. MoSHE/04/246/837/21). TASH and EKGH granted permission to perform the research. Material transfer agreement for shipping of samples to Uppsala university, Sweden, for further analyses was obtained from AAU, Ethiopia, 2021-08-17.

## 2.6. Statistics

IBM SPSS version 25.0 (Chicago, IL, USA) was used. Participants were characterized using descriptive summary metrics. The chi-square test was employed to examine whether categorical variables were related. Binary logistic regression was used to determine associations among variables age, sex, and co-morbidities with disease severity or mortality. To test the normal distribution of results, the Shapiro-Wilk test was used. Receiver operating characteristics (ROC) was used to assess the ability of NLR point-of-care at admission to predict disease severity or mortality. Values are given as mean ± SEM or median with IQR as appropriate. *p<0.05* or less was considered statistically significant.

# 3. Results

## 3.1. Demographic and clinical characteristics of patients

We studied 126 COVID-19 patients, separated into two groups: 68 had severe disease and 58 had a mild or moderate disease. The demographics of the groups are shown in **Table 1**.

   Almost three-fourth 91/126 (72%) of COVID-19 patients had either single or multiple morbidities such as DM (20/91 [22%]), hypertension (17/91 [19%]), CVD (21/91 [23%]), cancer (21/91 [23%]) or COPD (12/91 [13%]) (**Fig 1**).

## 3.2. Point-of-care biomarkers associated with severity and mortality

Clinical chemistry results showed that more than one-seventh 12/68 (18%) of patients in the severe group and 9/58 (16%) of patients in the mild/moderate group had polycythemia; more than one-fourth 20/68 (29%) of patients in the severe group and 27/58 (47%) in the mild/moderate group were anemic. The severe group of patients had deranged WBC with 12/68 (18%)

**Table 1. Demographics characteristic of COVID-19 patients.**

| | Age, Years, Median (IQR) | Gender, Male (%) |
|---|---|---|
| **COVID patients, n = 126** | 50 (20–86) | 72 (57.1) |
| **Mild/moderate disease, n = 58** | 32 (20–78) | 33 (56.9) |
| **Severe disease, n = 68** | 60 (22–86) | 39 (57.4) |
| **P-value** | **< .001[a]** | .959[a*] |

[*]Chi-square test

[a] comparison between mild or moderate and severe groups. Bold *p*-value emphasizes <0.05 criterion met. IQR, interquartile range

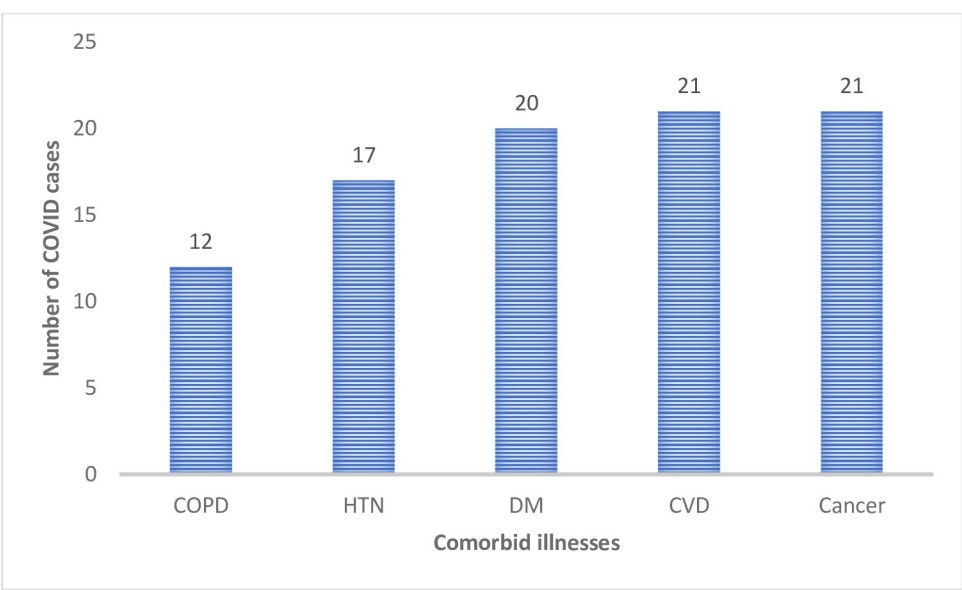

**Fig 1. Co-morbidity among COVID-19 patients admitted to the hospital.** CVD, Cardiovascular disease, COPD, Chronic obstructive pulmonary disease, DM, Diabetes Mellitus, HTN, Hypertension.

showing leukopenia and 14/68 (21%) with leukocytosis. Thrombocytopenia was more common among severely ill patients than in mild or moderate patients ($p < .05$) (**Table 2**).

The median NLR was significantly higher in the group presenting severe disease compared to those with mild or moderate disease. In addition, serum creatinine was more elevated among severe patients than in those with mild or moderate disease. Although, urea was more elevated among severe patients than mild or moderately ill patients, it was not statistically significant ($p > .05$) (**Table 2**).

The over-all mortality rate in our study was 15/126 (12%). There was a high mortality rate among individuals greater than 40 years of age [11/71 (16%)] and 7/54(13%) of the females. Disease complications were mainly seen in the severe group with 13 deaths reported. Severely ill patients were more predisposed to death than the mild or moderately ill patients ($p < .05$). We found 5/27 (19%) of non-surviving patients to have leukocytosis. Consequently, NLR values were significantly higher among non-survivors compared to survivors 15.2 (5.1–22.5) versus 4.26 (2.4–7.9), $p < .05$ (**Table 3**).

### 3.3. Predictive factors for disease severity and mortality

After adjusting for covariates, the odds of severe disease among patients who were 40 years and older was 4.86 times compared with those less than 40 years of age (AOR = 4.86,95% CI = 2.25,10.48, $p < .05$). Once being adjusted for other factors, the mortality odds in severely ill patients was 6.58-fold compared with those with mild/moderate disease (AOR = 6.58,95% CI = 1.29,33.56, $p < .05$). We found NLR to be a modest determinant and marker for disease outcome. After adjustment for other factors, NLR was associated with 1.07-fold (AOR = 1.07,95% CI = 1.02,1.12, $p < .05$) higher odds of dying compared to those with normal values for this marker (**Table 4**).

Receiver operating characteristics analysis revealed NLR to be a modest predictor of disease severity (sensitivity 69.1%, specificity 60.3%) at optimal cut-off >4.08, as well as prognostic marker for mortality risk (sensitivity 86.7%, specificity 55.9%) at optimal cut-off >4.63 (**Table 5, Fig 2A and 2B**).

**Table 2. Laboratory biomarker-related variables by COVID severity classification (n = 126).**

| Variable | COVID-19 disease severity | | |
|---|---|---|---|
| | Mild/moderate, (n = 58 (%) | Severe, (n = 68 (%) | P-value |
| **Hematocrit (%)** | | | .130 |
| <36 | 27(46.6) | 20(29.4) | |
| 36–45 | 22(37.9) | 36(52.9) | |
| >45 | 9(15.5) | 12(17.6) | |
| **White blood cell count ($10^3$/μL)** | | | .588 |
| <4.5 | 13 (22.4) | 14 (20.6) | |
| 4.5–11 | 31 (53.4) | 42 (61.8) | |
| >11 | 14 (24.1) | 12 (17.6) | |
| **Platelet count ($10^3$/μL)** | | | **.002** |
| <150 | 9(15.5) | 24(35.3) | |
| 150–450 | 40(69) | 43(63.2) | |
| >450 | 9(15.5) | 1(1.5) | |
| **Urea (mg/dL)** | | | .221 |
| <10 | 7(12.1) | 10(14.7) | |
| 10–45 | 47(81.0) | 47(69.1) | |
| >45 | 4(6.9) | 11(16.2) | |
| **Creatinine (mg/dL)** | | | **0.001** |
| <0.5 | 9 (15.5) | 3 (4.4) | |
| 0.5–0.9 | 38 (65.5) | 31 (45.6) | |
| >0.9 | 11 (19) | 34 (50) | |
| **ALT(U/L)** | | | .914 |
| = <33 | 31(53.4) | 37(54.4) | |
| >33 | 27(46.6) | 31(45.6) | |
| **AST (U/L) [fe]** | | | .337 |
| <10 | 2 (3.4) | 0 (0.0) | |
| 10–35 | 25 (43.1) | 29 (42.6) | |
| >35 | 31 (53.4) | 39 (57.4) | |
| **ALP(U/L)** | | | .597 |
| <45 | 4(6.9) | 7(10.3) | |
| 45–87 | 21(36.2) | 28(41.2) | |
| >87 | 33(56.9) | 33(48.5) | |
| **Neutrophil:Lymphocyte ratio [np]** | 3.23(2.09–5.39) | 6.68(3.03–12.21) | **.001** |

fe, fisher exact test; np, non-parametric (median with interquartile range)

The predictive value of creatinine for disease severity and mortality in COVID-19 patients was examined using a ROC curve analysis (**Fig 3A and 3B**). The AUC for illness severity and mortality were 0.68 and 0.76, respectively. Creatinine showed a sensitivity of 80% and a specificity of 70% in predicting mortality in COVID-19 (**Table 5**).

## 4. Discussion

Faced with the threat of a COVID-19 pandemic, significant efforts have been made to find clinical and laboratory prognostic indicators that can aid in triage and medical health resource allocation. NLR is one such indicator that has been thoroughly studied and is considered as a good predictive potential using varying cutoffs [18]. In this investigation, we assessed demographic profiles (age, gender), laboratory biomarkers (NLR), organ function biomarkers (renal

**Table 3. Demographic, clinical and laboratory profiles by survivorships status.**

| Variable | COVID-19 survivorship status | | |
|---|---|---|---|
| | Non-survivor, n = 15 (%) | Survivor, n = 111(%) | P-value |
| **Age** | | | .158 |
| < = 40 | 4(7.3) | 51(92.7) | |
| >40 | 11(15.5) | 60(84.5) | |
| **Gender** | | | .751 |
| Male | 8(11.1) | 64(88.9) | |
| Female | 7(13.0) | 47(87.0) | |
| **Comorbid illness** | | | .169 |
| Yes | 10(15.9) | 53 (84.1) | |
| No | 5(7.9) | 58 (92.1) | |
| **Disease severity** | | | **.007** |
| Mild/moderate | 2(3.4) | 56 (96.6) | |
| Severe | 13(19.1) | 55 (80.9) | |
| **Hematocrit (%)** | | | .403 |
| <36 | 5 (10.6) | 42 (89.4) | |
| 36–45 | 9 (15.5) | 49 (84.5) | |
| >45 | 1 (4.8) | 20 (95.2) | |
| **White blood cell count ($10^3$/μL)** | | | .305 |
| <4.5 | 5 (18.5) | 22 (81.5) | |
| 4.5–11 | 6 (8.2) | 67 (91.8) | |
| >11 | 4 (15.4) | 22 (84.6) | |
| **Platelet count ($10^3$/μL)** | | | .473 |
| <150 | 4 (12.1) | 29 (87.9) | |
| 150–450 | 11 (13.3) | 72 (86.7) | |
| >450 | 0 (0) | 10 (100) | |
| **Urea (mg/dL) [fe]** | | | .129 |
| <10 | 1(5.9) | 16(94.1) | |
| 10–45 | 10(10.6) | 84 (89.4) | |
| >45 | 4(26.7) | 11(73.3) | |
| **Creatinine (mg/dL)** | | | **.001** |
| <0.5 | 0 (0.0) | 12 (100) | |
| 0.5–0.9 | 3 (4.3) | 66 (95.7) | |
| >0.9 | 12 (26.7) | 33 (73.3) | |
| **ALT(U/L)** | | | .958 |
| = <33 | 8 (11.8) | 60 (88.2) | |
| >33 | 7 (12.1) | 51 (87.9) | |
| **AST (U/L) [fe]** | | | .553 |
| <10 | 0 (0.0) | 2 (100) | |
| 10–35 | 5 (9.3) | 49 (90.7) | |
| >35 | 10 (14.3) | 60 (85.7) | |
| **ALP(U/L)** | | | .095 |
| <45 | 2 (18.2) | 9 (81.8) | |
| 45–87 | 2 (4.1) | 47 (95.9) | |
| >87 | 11 (16.7) | 55 (83.3) | |
| **Neutrophil:Lymphocyte ratio[np]** | 15.17(5.13–22.5) | 4.26(2.40–7.90) | **.002** |

fe, fisher exact test, np, non-parametric test (median with inter-quartile range)

**Table 4. Association of sociodemographic features, clinical history, laboratory profile with severity and mortality among COVID-19 patients (n = 126).**

| Variables | Severity | | Mortality | |
|---|---|---|---|---|
| | AOR (95%CI) | P-value | AOR (95%CI) | P-value |
| **Age** | | | | |
| >40 | 4.86(2.25,10.48) | **.001** | 1.23(.29, 5.26) | .779 |
| < = 40 | 1 | | 1 | |
| **Sex** | | | | |
| Male | .91(.42,1.98) | .807 | .59(.17, 2.00) | .394 |
| Female | 1 | | 1 | |
| **Comorbidity** | | | | |
| Yes | 1.18(.54,2.58) | .675 | 1.72(.50,5.88) | .389 |
| No | 1 | | 1 | |
| **Disease Severity** | NA | NA | | |
| Severe | | | 6.58(1.29,33.56) | **.023** |
| Mild/Moderate | | | 1 | |
| **Neutrophil:Lymphocyte ratio** | 1.04(.99,1.09) | .105 | 1.07(1.02,1.12) | **.004** |

NA, not applicable, Comorbidity (Cancer, CVD, COPD, DM and Hypertension)

function tests) and their association with clinical outcome (severity and mortality) of COVID-19 patients.

The average age of severely ill patients was considerably higher than mild/moderate ones, in line with earlier research [19]. Furthermore, after controlling for other factors, being 40 years of age or older was linked to a 4.9-fold greater risk of acquiring serious illness. Elderly patients were found to be more inclined to COVID-19 illness severity in many studies conducted in Ethiopia, China, the United States, and Europe, which is similar to the finding of the current study [3, 20–22]. The causes for this could include a higher risk of severity due to a weakened immune system and co-morbid disorders, rendering older people more susceptible to quickly progressing illnesses that would have a poor outcome. Our study population consisted mainly of male individuals with a male: female ratio of 1.33:1 where males were more likely to develop severe illness. In this context, our study supports previous results [23, 24]. However, one study found that once a severe condition has developed, the mortality risk of women is comparable to that of men [25].

Thrombocytopenia was significantly more common in severely ill patients than in mild or moderate cases. SARS-CoV-2 impairs megakaryocyte maturation and platelet formation due to viral affection on the hematopoiesis [26]. Patients with SARS and Middle East respiratory disease have also been documented to have thrombocytopenia [27], considered to be due to aberrant megakaryocyte maturation [28, 29].

**Table 5. Creatinine and NLR as predictor of disease severity and mortality of COVID-19.**

| | Test variables | | | |
|---|---|---|---|---|
| | NLR | | Creatinine | |
| | Disease Severity | Death | Disease Severity | Death |
| **Cut-off** | > 4.08 | > 4.63 | > 2.50 | > 2.50 |
| **Area (95%CI)** | .67 (.577-.768) | .75 (.597-.905) | .68 (.582-.770) | .76 (.644-.880) |
| **Sensitivity** | 69.1 | 86.7 | 50.0 | 80 |
| **Specificity** | 60.3 | 55.9 | 81 | 70.3 |
| **P-value** | **.001** | **.002** | **.001** | **.001** |

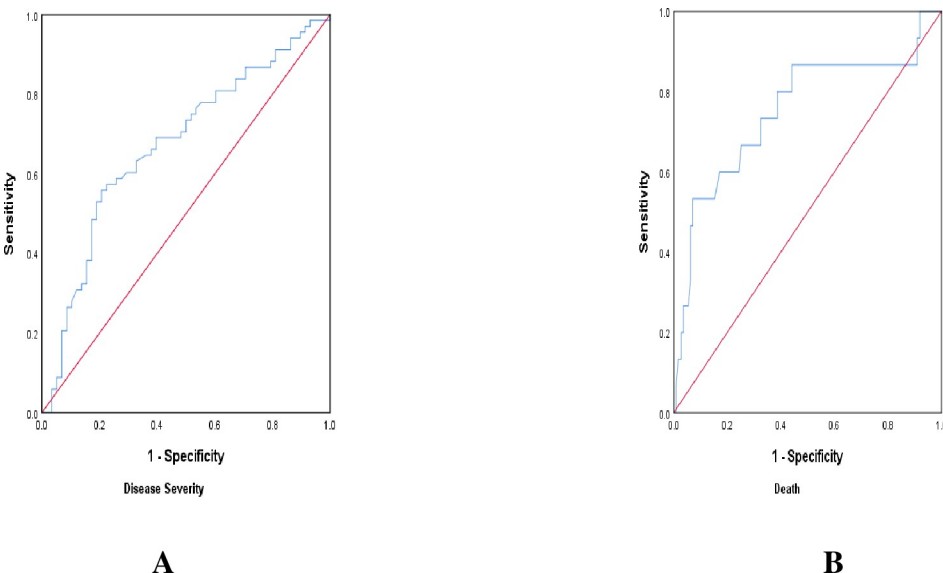

**Fig 2.** ROC curve analysis to predict NLR as predictor of disease severity (A) and mortality (B).

Two meta-analyses have confirmed relationships between greater NLR and COVID-19 severity and mortality in [30, 31]. Additionally, a Turkish study found that high NLR predisposes to mortality, suggesting that this point-of-care biomarker may be useful in predicting COVID-19 mortality [32]. Despite the research on NLR as a measure of disease severity in COVID-19, the cut-off values are highly variable, indicating limited usefulness of this biomarker for prediction of disease outcomes. It is also unclear what NLR ratios should be typical for a healthy adult. COVID-19 patients in Ethiopia had cut-off scores greater than 3.0 in studies that predict severe illness [22]. In our current investigation, however, the cut-off value for

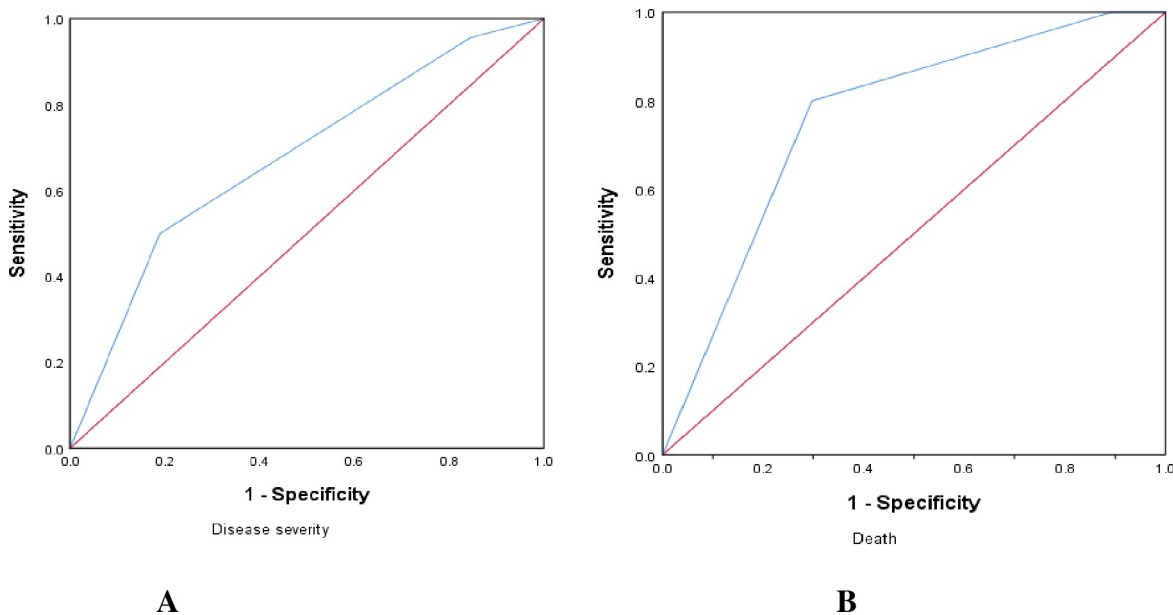

**Fig 3.** ROC curve analysis to predict creatinine as predictor of disease severity (A) and mortality (B).

NLR in predicting severe illness was more than 4.08. Furthermore, NLR greater than 4.63 was associated with only a 1.07-fold increased risk of death as compared to those having a NLR less than 4.00, which suggests a limited value of the predictability of this biomarker. Increased NLR is associated with increased systemic inflammatory hyperactivation which could exacerbate a cytokine storm leading to tissue damage [33]. In patients with comorbidities, NLR may maintain its capacity to predict COVID-19 severity. For example, in hospitalized patients with various types of malignancies, NLR has been implicated to predict COVID-19 severity and survival [34]. This is also consistent with research findings in China and reports from systematic reviews where NLR was found to an important predictor of disease severity and outcome [22, 31, 35, 36].

The diagnostic performance of the ROC curve for NLR as regards disease severity was 0.672 and mortality 0.751, which revealed that NLR should be of limited value for early categorization of the severity of COVID-19.

Serum creatinine was substantially higher in severely ill patients than in mild/moderate patients and serum urea was likewise higher. Increased serum creatinine and urea values may indicate abnormal renal function in COVID-19 but might as well indicate low glomerular filtration due to cardiac failure [37–39]. The angiotensin-converting enzyme 2 (ACE2) is a widely accepted receptor of SARS-CoV-2 [40]. When the virus binds to ACE2, it prevents ACE2 from performing its normal function resulting in reduced renal perfusion and filtration, that is why serum creatinine and urea rise. To this end kidney tubular cells, which express the ACE2 receptor on their cellular surface, could be directly affected by SARS-CoV-2 [41]. Evidence also shows that kidney-resident cells can interact with circulating mediators, resulting in microcirculatory derangement, endothelial dysfunction and tubular injury [42, 43]. In support of this, patients with renal failure are more likely to develop acute kidney injury when infected with COVID-19 as shown in a retrospective single-center study [44]. In our current study, non-survivors had a 26.7% increase of creatinine. According to reports, 25–30% of people infected with SARS-CoV-2 develop acute kidney injury, which has been linked to an increased mortality risk [45, 46]. Moreover, a meta-analysis covering 41 studies found kidney disease to be strongly linked to the severity and mortality of COVID-19 [47].

This study has some limitations. The study did not further explore the potential impact of COPD, CVD and other illnesses on study variables due to the minimal number of particular diseases in comorbidities. Instead, the combined impact of comorbidities was evaluated. We believe the study offers significant insights on clinical characteristics of this recently emerging disease, COVID-19, despite these limitations.

## 5. Conclusion

In patients hospitalized with COVID-19, NLR and serum creatinine were higher in severe cases compared to mild/moderate. On-admission, point-of-care NLR may predict COVID-19 and be utilized as a risk stratification tool because it is a standard, accessible and cost-effective measurement. A NLR cut-off of 4.63 was found to distinguish between non-survivor and survivor outcomes. Thus, the usefulness of NLR as a predictor of COVID-19 severity and mortality is only modest and of limited value in the clinical setting.

## Supporting information

**S1 Data.**
(SAV)

## Acknowledgments

A.T sincerely acknowledges support from CDT-Africa of Addis Ababa University and Professor Per M. Hellström and Associate professor Dominic-Luc Webb, Department of Medical Sciences, Gastroenterology and Hepatology Unit, Uppsala University, Sweden.

## Author Contributions

**Conceptualization:** Abdisa Tufa, Dominic-Luc Webb, Per M. Hellström, Solomon Genet.

**Data curation:** Abdisa Tufa, Tewodros Getinet, Tsegaye Gebreyes Hundie, Dominic-Luc Webb, Per M. Hellström, Solomon Genet.

**Formal analysis:** Yidnekachew Asrat, Tewodros Getinet, Tsegaye Gebreyes Hundie, Solomon Genet.

**Funding acquisition:** Tsegahun Manyazewal, Dominic-Luc Webb, Per M. Hellström, Solomon Genet.

**Investigation:** Abdisa Tufa, Tewodros Haile Gebremariam, Yidnekachew Asrat, Dominic-Luc Webb, Per M. Hellström, Solomon Genet.

**Methodology:** Abdisa Tufa, Tewodros Haile Gebremariam, Tsegahun Manyazewal, Dominic-Luc Webb, Per M. Hellström, Solomon Genet.

**Software:** Tewodros Getinet.

**Supervision:** Tewodros Haile Gebremariam, Tsegahun Manyazewal, Tsegaye Gebreyes Hundie, Dominic-Luc Webb, Per M. Hellström.

**Writing – original draft:** Abdisa Tufa, Dominic-Luc Webb.

**Writing – review & editing:** Tewodros Haile Gebremariam, Tsegahun Manyazewal, Yidnekachew Asrat, Tewodros Getinet, Tsegaye Gebreyes Hundie, Dominic-Luc Webb, Per M. Hellström, Solomon Genet.

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
