## [Decision Letter · Decision Letter 0]

14 Jul 2022

PONE-D-22-15252Renal function biomarkers and neutrophil to lymphocyte ratio as a best predictor of disease severity and mortality among hospitalized patients with COVID-19PLOS ONE

Dear Dr. Tufa,

Thank you for submitting your manuscript to PLOS ONE. After careful consideration, we feel that it has merit but does not fully meet PLOS ONE’s publication criteria as it currently stands. Therefore, we invite you to submit a revised version of the manuscript that addresses the points raised during the review process.

We look forward to receiving your revised manuscript.

Kind regards,

Tai-Heng Chen, M.D.

Academic Editor

PLOS ONE

Journal Requirements:

All authors declare that they have no known competing financial interests or personal relationships that could have appeared to influence the work reported herein.

 No authors received funding for this particular study. 

Reviewers' comments:

Reviewer's Responses to Questions

**Comments to the Author**

1. Is the manuscript technically sound, and do the data support the conclusions?

Reviewer #1: Partly

Reviewer #2: Yes

2. Has the statistical analysis been performed appropriately and rigorously? 

Reviewer #1: Yes

Reviewer #2: Yes

3. Have the authors made all data underlying the findings in their manuscript fully available?

Reviewer #1: Yes

Reviewer #2: Yes

4. Is the manuscript presented in an intelligible fashion and written in standard English?

Reviewer #1: Yes

Reviewer #2: Yes

5. Review Comments to the Author

Reviewer #1: The manuscript entitled ""Renal function biomarkers act as a best predictor of disease severity and mortality among hospitalized patients with COVID-19" reported that high creatinine levels, as well as NLR related to disease severity and mortality in patients with COVID-19. ROC curve analysis was used to access the predictive value of disease severity and mortality for NLR. Some comments are listed below:

1. The predictive value of creatinine levels is unrevealed in this manuscript, it is not consistent with the conclusion.

2. Based on the ROC curve analysis of NLR, the AUC was only 0.672 for disease severity, with sensitivity of 69.1% and specificity of 60.3%. These results only suggested the ability of NLR to predict disease severity, but not "the most precise biomarker for predicting severity" or "as a best predictor of disease severity" that described in manuscript.

3. Other than renal failure, whether other diseases that listed in this manuscript, such as COPD abd Cardiovascular diseases could affect the severity of Covid-19 infection?

4. Inclusion/exclusion criteria, diagnostic details should be added in materials and methods part.

Reviewer #2: Dear authors,

You investigated the role of NLR in COVID-19 which is a simple and practical biomarker that can predict mortality in COVID-19 patients. It would be more appropriate if you can cite the below article.

Ergenç H, Ergenç Z, Dog An M, Usanmaz M, Gozdas HT. C-reactive protein and neutrophil-lymphocyte ratio as predictors of mortality in coronavirus disease 2019. Rev Assoc Med Bras (1992). 2021 Oct;67(10):1498-1502. doi: 10.1590/1806-9282.20210679.

6. PLOS authors have the option to publish the peer review history of their article (what does this mean?). If published, this will include your full peer review and any attached files.

Reviewer #1: No

Reviewer #2: **Yes: **Hasan Tahsin Gozdas

---

## [Author Response · Author response to Decision Letter 0]

11 Aug 2022

Tai-Heng Chen, M.D.

Academic Editor

PLOS ONE

Re: Revision of PONE-D-22-15252 entitled “Neutrophil-to-lymphocyte ratio and serum creatinine as predictors of disease severity and mortality in patients hospitalized with COVID-19”

Dear Dr. Chen,

 We thank the editorial team and the reviewers for their very important comments. We have made careful revisions to the original manuscript based on these comments. In the attached letter, we have responded point-by-point to each of the comments that arose during the peer review process. Our responses are in italics against numbered reviewer comments written in plain text. 

Sincerely, 

Abdisa Tufa

Addis Ababa University, Ethiopia

On behalf of the Authors

Response to Reviewer 1: 

1. The predictive value of creatinine levels is unrevealed in this manuscript, it is not consistent with the conclusion.

Response: 

We agree. The predictive value of serum creatinine is incorporated into the manuscript text, figure and table [ Text (line 193-196), Figure 3A and 3B (line 198-199) and table 5 (line 200)]. We have now revised the Conclusion and some aspects of the Result sections.

2. Based on the ROC curve analysis of NLR, the AUC was only 0.672 for disease severity, with sensitivity of 69.1% and specificity of 60.3%. These results only suggested the ability of NLR to predict disease severity, but not "the most precise biomarker for predicting severity" or "as a best predictor of disease severity" that described in manuscript. 

Response: 

We agree and have now carefully revised the Abstract and Main Text to assure that NLR had modest sensitivity and specificity for predicting disease severity and mortality. (indicated in lines 45-46 and 242-243). 

We also modified the title of the manuscript to address the concern, thus revising it as ’Neutrophil-to-lymphocyte ratio and serum creatinine as predictors of disease severity and mortality in patients hospitalized with COVID-19” (line 1-3). 

3. Other than renal failure, whether other diseases that listed in this manuscript, such as COPD and Cardiovascular diseases could affect the severity of Covid-19 infection? 

Response:

We reported that comorbidities had no clear impact on the severity of COVID-19 or mortality (table 4, line 187-189). Due to the low number of specific diseases among comorbidities, the study did not further investigate the potential effects of COPD, CVD, and other illnesses on study variables. Now, we have added this to the Discussion section as a limitation (line 263-267). 

4. Inclusion/exclusion criteria, diagnostic details should be added in materials and methods part.

Response: 

We agree and have now included the details of the inclusion/exclusion criteria in the manuscript, - ‘2.2. Study subjects’ in the materials and methods part. (Line 90-98). 

 Response to reviewer 2:

1. You investigated the role of NLR in COVID-19 which is a simple and practical biomarker that can predict mortality in COVID-19 patients.

Response: We thank the Reviewer for the comment.

2. It would be more appropriate if you can cite the below article

Response: Thank you for sharing this important article that we read and cited in the current manuscript accordingly (Line 226-228, reference 32, line 385-387). 

Journal requirements

 Response: Thank you for letting us know the specific PLOS ONE requirements that should be addressed. With this we have now:

- Made necessary revisions to the manuscript to meet PLOS ONE's style. 

- Included details regarding participant consent.

- Included Financial Disclosure statement, thus “No authors received funding for this particular study. Centre for Innovative Drug Development and Therapeutic Trials for Africa (CDT-Africa), College of Health Sciences, Addis Ababa University, Addis Ababa supported expenses associated with data and sample collection. Professor Per M. Hellström and Associate professor Dominic-Luc Webb, Gastroenterology and Hepatology Unit, Department of Medical Sciences, Uppsala University, Sweden support all expenses related to laboratory analysis. The funders have no role in the study design, data analysis and decision to publish including preparation of the manuscript”.

- Included Competing Interests, thus “The authors, Abdisa Tufa, Dr. Solomon Genet, Dr. Tewodros Haile Gebremariam and Dr. Yidnekachew Asrat receive their salary from Addis Ababa University. Dr. Tsegahun Manyazewal receives salary from CDT-Africa, Dr. Tsegaye Gebreyes receives salary from Eka Kotobe General Hospital. Tewodros Getinet receives salary from St. Paul’s Hospital Millennium Medical College. Professor Per M. Hellström and Dr. Dominic-Luc Webb receive their salary from Uppsala University”. All authors declare that they have no known competing financial interests or personal relationships that could have appeared to influence the work reported herein.

---

## [Decision Letter · Decision Letter 1]

1 Sep 2022

PONE-D-22-15252R1Neutrophil -to- lymphocyte ratio and serum creatinine as predictors of disease severity and mortality in patients hospitalized with COVID-19PLOS ONE

Dear Dr. Tufa,

Thank you for submitting your manuscript to PLOS ONE. After careful consideration, we feel that it has merit but does not fully meet PLOS ONE’s publication criteria as it currently stands. Therefore, we invite you to submit a revised version of the manuscript that addresses the points raised during the review process.

We look forward to receiving your revised manuscript.

Kind regards,

Tai-Heng Chen, M.D.

Academic Editor

PLOS ONE

Journal Requirements:

Reviewers' comments:

Reviewer's Responses to Questions

**Comments to the Author**

1. If the authors have adequately addressed your comments raised in a previous round of review and you feel that this manuscript is now acceptable for publication, you may indicate that here to bypass the “Comments to the Author” section, enter your conflict of interest statement in the “Confidential to Editor” section, and submit your "Accept" recommendation.

Reviewer #1: (No Response)

Reviewer #2: All comments have been addressed

2. Is the manuscript technically sound, and do the data support the conclusions?

Reviewer #1: Partly

Reviewer #2: Yes

3. Has the statistical analysis been performed appropriately and rigorously? 

Reviewer #1: Yes

Reviewer #2: Yes

4. Have the authors made all data underlying the findings in their manuscript fully available?

Reviewer #1: Yes

Reviewer #2: Yes

5. Is the manuscript presented in an intelligible fashion and written in standard English?

Reviewer #1: Yes

Reviewer #2: Yes

6. Review Comments to the Author

Reviewer #1: Due to the insufficient evidence in the predictive abilities of both N/L ratio and glucose creatinine, their predictive value s in severity and mortality of Covid-19 patients should be seriously considered, as well as the title of revised manuscript.

Reviewer #2: (No Response)

7. PLOS authors have the option to publish the peer review history of their article (what does this mean?). If published, this will include your full peer review and any attached files.

Reviewer #1: No

Reviewer #2: **Yes: **Hasan Tahsin Gozdas

---

## [Author Response · Author response to Decision Letter 1]

7 Sep 2022

Tai-Heng Chen, M.D.

Academic Editor

PLOS ONE

Re: Revision of PONE-D-22-15252R1 entitled “Limited value of neutrophil-to-lymphocyte ratio and serum creatinine as point-of-care biomarkers of disease severity and infection mortality in patients hospitalized with COVID-19” 

Dear Dr. Chen,

 We appreciate the editorial staffs and the reviewers' insightful comments. Based on these remarks, we carefully revised the original manuscript. We have addressed each of the criticisms that were made throughout the peer review procedure in detail in the accompanying letter. In contrast to the plain text, numbered reviewer comments, our responses are printed in red ink. 

Sincerely, 

Abdisa Tufa

Addis Ababa University, Ethiopia

On behalf of the Authors

Response to Reviewer 1: 

1. Reviewer #1: Due to the insufficient evidence in the predictive abilities of both N/L ratio and glucose creatinine, their predictive value s in severity and mortality of Covid-19 patients should be seriously considered, as well as the title of revised manuscript.

REPLY: The title has been re-evaluated to give a more humble feature to our findings with the NLR measurements, which seem to be of limited value as studied in our patient cohorts. Likewise, the conclusion has been rectified and the usefullness of the NLR value clearly stated to be of limited value for the prediction of the disease prognosis. For clarity, these requested changes have been underlined.

We have gone through the text body in order to make the whole message of the result section to be in line with conclusion as presented against the background prevailing research data in the discussion.

Journal requirements

Please review your reference list to ensure that it is complete and correct.

 Reply: We have checked the lists of references it is complete and correct.

---

## [Decision Letter · Decision Letter 2]

15 Sep 2022

Limited value of neutrophil-to-lymphocyte ratio and serum creatinine as point-of-care biomarkers of disease severity and infection mortality in patients hospitalized with COVID-19

PONE-D-22-15252R2

Dear Dr. Tufa,

We’re pleased to inform you that your manuscript has been judged scientifically suitable for publication and will be formally accepted for publication once it meets all outstanding technical requirements.

Kind regards,

Tai-Heng Chen, M.D.

Academic Editor

PLOS ONE

Reviewers' comments:

Reviewer's Responses to Questions

**Comments to the Author**

1. If the authors have adequately addressed your comments raised in a previous round of review and you feel that this manuscript is now acceptable for publication, you may indicate that here to bypass the “Comments to the Author” section, enter your conflict of interest statement in the “Confidential to Editor” section, and submit your "Accept" recommendation.

Reviewer #1: All comments have been addressed

2. Is the manuscript technically sound, and do the data support the conclusions?

Reviewer #1: Yes

3. Has the statistical analysis been performed appropriately and rigorously? 

Reviewer #1: Yes

4. Have the authors made all data underlying the findings in their manuscript fully available?

Reviewer #1: Yes

5. Is the manuscript presented in an intelligible fashion and written in standard English?

Reviewer #1: Yes

6. Review Comments to the Author

Reviewer #1: (No Response)

7. PLOS authors have the option to publish the peer review history of their article (what does this mean?). If published, this will include your full peer review and any attached files.

Reviewer #1: No

---

## [Editor Report · Acceptance letter]

26 Sep 2022

PONE-D-22-15252R2 

Limited value of neutrophil-to-lymphocyte ratio and serum creatinine as point-of-care biomarkers of disease severity and infection mortality in patients hospitalized with COVID-19 

Dear Dr. Tufa:

I'm pleased to inform you that your manuscript has been deemed suitable for publication in PLOS ONE. Congratulations! Your manuscript is now with our production department. 

Kind regards, 

on behalf of

Dr. Tai-Heng Chen 

Academic Editor

PLOS ONE